# Assessment of the Molecular Heterogeneity of E-Cadherin Expression in Invasive Lobular Breast Cancer

**DOI:** 10.3390/cancers14020295

**Published:** 2022-01-07

**Authors:** John Alexander, Odette Mariani, Celine Meaudre, Laetitia Fuhrmann, Hui Xiao, Kalnisha Naidoo, Andrea Gillespie, Ioannis Roxanis, Anne Vincent-Salomon, Syed Haider, Rachael Natrajan

**Affiliations:** 1The Breast Cancer Now Toby Robins Research Centre, The Institute of Cancer Research, London SW3 6JB, UK; john.alexander@icr.ac.uk (J.A.); hui.xiao@icr.ac.uk (H.X.); kalnisha.naidoo1@nhs.net (K.N.); andrea.gillespie@icr.ac.uk (A.G.); Ioannis.Roxanis@icr.ac.uk (I.R.); 2Department of Diagnostic and Theranostic Medicine, Department of Pathology, Institute Curie, PSL-Research University, 75005 Paris, France; odette.mariani@curie.fr (O.M.); celine.meaudre@curie.fr (C.M.); laetitia.fuhrmann@curie.fr (L.F.); anne.salomon@curie.fr (A.V.-S.); 3Department of Cellular Pathology, King’s College Hospital, Denmark Hill, London SE5 9RS, UK

**Keywords:** invasive lobular breast cancer, heterogeneity, E-cadherin

## Abstract

**Simple Summary:**

Invasive lobular breast cancers (ILCs) are histologically classified by their discohesive growth pattern, due to loss of the cell adhesion glycoprotein E-cadherin (CDH1), which arises via mutation in *CDH1* in around half of these tumours. A subset of these tumours, however, show mixed levels of E-cadherin expression. Here, we sought to address whether the distinct parts of individual tumours showing heterogeneous E-cadherin expression harbour distinct driver alterations. Using whole genome sequencing and methylation profiling of nine such cases, we identified that these tumours are clonally related, suggesting that they are part of the spectrum of ILC tumours. *CDH1* mutant tumours showed a higher mutational burden indicative of APOBEC-mediated mutagenesis. In some cases, known clinically actionable driver mutations, such as *PIK3CA,* were exclusive to one component. Together, these results highlight the heterogeneity underpinning this special histological breast cancer.

**Abstract:**

Mutations and loss of E-cadherin protein expression define the vast majority of invasive lobular carcinomas. In a subset of these cases, the heterogeneous expression of E-cadherin is observed either as wild-type (strong membranous) expression or aberrant expression (cytoplasmic expression). However, it is unclear as to whether the two components would be driven by distinct genetic or epigenetic alterations. Here, we used whole genome DNA sequencing and methylation array profiling of two separately dissected components of nine invasive lobular carcinomas with heterogeneous E-cadherin expression. E-cadherin negative and aberrant/positive components of E-cadherin heterogeneous tumours showed a similar mutational, copy number and promoter methylation repertoire, suggesting they arise from a common ancestor, as opposed to the collision of two independent tumours. We found that the majority of E-cadherin heterogeneous tumours harboured *CDH1* mutations in both the E-cadherin negative and aberrant/positive components together with somatic mutations in additional driver genes known to be enriched in both pure invasive carcinomas of no special type and invasive lobular breast cancers, whereas these were less commonly observed in *CDH1* wild-type tumours. *CDH1* mutant tumours also exhibited a higher mutation burden as well as increased presence of APOBEC-dependent mutational signatures 2 and 13 compared to *CDH1* wild-type tumours. Together, our results suggest that regardless of E-cadherin protein expression, tumours showing heterogeneous expression of E-cadherin should be considered as part of the spectrum of invasive lobular breast cancers.

## 1. Introduction

Invasive lobular breast cancer (ILC) is the most common of the special histological types of breast cancer, accounting for up to 10–15% of all breast cancers diagnosed. Compared to invasive carcinomas of no special type (IC-NST), ILCs are characterised by special growth patterns and display discohesive cells individually dispersed or arranged in single files linearly in fibrous stroma [1]. ILC differs from IC-NST in its epidemiology, molecular alterations, clinicopathologic features and responsiveness to therapy [2,3,4,5]. At the molecular level, the characteristic discohesive hallmark of ILC is due to the dysregulation of cell–cell adhesion, primarily driven by a lack of E-cadherin (CDH1) protein expression observed in the majority of cases and is a discriminating feature of ILC, as strong membranous expression of E-cadherin is observed in the majority of IC-NSTs [2]. Thus, loss of E-cadherin in ILC is thought to contribute to its pathogenesis and different metastatic patterns, in particular, the higher prevalence of metastasis to serosal surfaces, such as the ovary and peritoneum [1,2]. Molecularly, the patterns of E-cadherin loss follow a classic Knudsen’s two-hit tumour suppressor hypothesis, involving *CDH1* mutation (50–60% cases) [6], gene methylation (21–77% of cases) [7,8] and/or loss of heterozygosity in the region of 16q22.1 (or whole chromosomal arm) [2]. These molecular events occur early on in tumorigenesis and are often seen in lobular carcinoma in situ (LCIS), suggesting that these tumours show an early pattern of evolution [9,10], and are rarely seen in IC-NST [6] making them a discriminatory feature of ILC.

E-cadherin loss in ILC also results in the loss of α-, β- and γ-catenins, and upregulation and relocation to the cytoplasm of p120-catenin, which enables anoikis resistance through the activation of Rho/Rock signalling, allowing cell survival in the absence of adjacent epithelial cells [11,12,13]. Thus, E-cadherin expression has become an important diagnostic feature ILC and is being increasingly used as a tool to differentiate between lobular and IC-NST lesions in some diagnostic situations. However, approximately 10% of ILCs still express E-cadherin [10,13], either with normal membrane localisation or aberrantly distributed as fragmented membrane and/or cytoplasmic staining. In these cases, the E-cadherin–catenin complex may also be dysfunctional, due to the presence of the *CDH1* gene mutation or aberrant/loss of expression of the catenin binding proteins [14]. 

Amongst the spectrum of ILC tumours, 3–5% show a heterogeneous E-cadherin phenotype, either showing mixed histology with both ILC and IC-NST components or ILC histology with differential E-cadherin expression, i.e., complete loss and either weak or aberrant cytoplasmic expression [15]. Recent evidence suggests a model whereby separate morphological components of these mixed tumours can arise from a common ‘ductal’ ancestor. In particular, mixed IC-NST and ILC tumours (MDLs) that present with LCIS and DCIS show an early clonal divergence associated with complete loss of E-cadherin expression, whist in the majority of MDLs, which present with DCIS but not LCIS, direct clonal divergence from ductal to the lobular phenotype occurs late in tumour evolution and is associated with aberrant expression of E-cadherin [9]. Furthermore, somatic profiling using exome sequencing of a small cohort (*n* = 4) of MDLs has suggested that *CDH1* and *ESR1* mutations are associated with clonal divergence in MDL [9]. Recent evidence from the TCGA breast cancer study has additionally highlighted that lobular and NST carcinomas themselves harbour distinct genomic alterations aside from loss of E-cadherin, including a higher frequency of *FOXA1* mutations and AKT pathway activation, suggesting different mechanisms of pathway modulation in these tumours [6]. Importantly, a recent seminal study using synthetic–lethal approaches in ILC breast cancer cell models with complete loss of E-cadherin expression suggested a specific therapeutic vulnerability to ROS1 inhibitors, providing the preclinical rationale for assessing ROS1 inhibitors, such as the licensed drug crizotinib, in ILCs with absent E-cadherin expression [16].

Based on the above, we hypothesised that distinct components of invasive lobular breast cancers with heterogeneous E-cadherin expression may harbour different genomic alterations and that the resultant pathway activation and, hence, subsequent clinical response to potential targeted agents, such as crizotinib, may be heterogeneous. To address this, we set out to study the genomic alterations in a cohort of nine E-cadherin heterogeneous ILCs by dissecting the different morphological components of diagnosed invasive lobular breast cancers showing absent and aberrant or intact E-cadherin expression. Here, we aimed to determine (i) the somatic copy number and mutational repertoire of E-cadherin negative and aberrant/positive components of E-cadherin heterogeneous ILCs; (ii) the methylation repertoire of E-cadherin negative and aberrant/positive components of E-cadherin heterogeneous ILCs; (iii) additional potential driver genetic alterations of the E-cadherin-negative and aberrant/positive components of E-cadherin heterogeneous ILCs. Overall, we show that the negative and aberrant components of these E-cadherin heterogeneous tumours arise from a common ancestor and that phenotypic diversity is likely due to clonal progression. Differences in targetable driver alterations were observed in a small number of cases, highlighting the potential importance of assessment of these in the clinical management of these patients.

## 2. Methods

### 2.1. Cases and Histological Review

Eighteen cases diagnosed as classical invasive lobular breast cancer displaying heterogeneous E-cadherin expression were retrieved from the Institut Curie tissue bank. Diagnostic slides were reviewed by at least two consultant breast pathologists (AVS, IR and KN). Representative sections of each case used for DNA extraction were reviewed, and the tumour cell content and composition of the areas displaying absent or aberrant E-cadherin expression were estimated. Samples were anonymised prior to analysis. This study was approved by the local Institutional Review Boards of the authors’ institutions (ethical approval number: 2016037) (Appendix A).

### 2.2. Immunohistochemistry

We used primary antibodies against E-cadherin (CDH1) (4A2C7 (ThermoFisher, Invitrogen, # 33-4000), dilution 1/100 (5 µg/mL), Beta-catenin clone 14 (BD, 610 154), dilution: 1/200, P120/delta-catenin: clone D7S2M (Cell Signalling, 598 54S), dilution: 1/400 with an incubation time of 30 min. All IHC was performed using the Menarini automate Impath36 with the kit “IMPATH DAB OB DETECTION KIT” (Menarini Diagostics, U.K.). Staining was visualised using 3,3’-diaminobenzidine (DAB) secondary antibody.

### 2.3. Macrodissection and DNA Extraction

Three-micrometre-thick representative sections of the snap-frozen blocks of histologically confirmed classical ILCs with heterogeneous E-cadherin expression (EcadhetILC) were stained with E-cadherin to provide a reference. Cases with clearly separated components were macrodissected from the frozen tissue block with a scalpel to recover the tumour fragment and to ensure enrichment of the tumour cell content as previously described [17]. Matched germline DNA was extracted from peripheral blood as previously described [18] or macrodissected from adjacent normal breast tissue for each case. To avoid the possibility of morphologically appearing non-neoplastic cells harbouring somatic mutations, we prioritised the dissection of stromal cells and avoided normal breast ducts and lobules. DNA was extracted using the DNeasy Blood and Tissue Kit (Qiagen, Germany), according to the manufacturers’ guidelines. DNA quantity and quality were analysed with the Agilent Bioanalyzer (Agilent) on 0.8% agarose gel with SYBR^™^ Safe DNA Gel Stain (Invitrogen).

### 2.4. Whole Genome Sequencing

#### 2.4.1. Library Construction 

Tumour tissues and matched germline were subjected to whole-genome sequencing using 1 μg of genomic DNA. All constructed libraries were loaded onto BGISEQ-500 (MGI Tech Co., Ltd., Shenzhen, China), and the sequences were generated as 100 bp paired-end reads as previously described [19,20]. Fragments were end repaired and then 3′ adenylated for adaptor ligation. Then, they were washed with Ampure XP beads. Seven PCR cycles were performed, using primers complementary to the ligated adapters. The double-stranded PCR products were heat denatured and circularised by the splint oligo sequence, and single strand circle DNA (ssCir DNA) were formatted as the final library. The library was amplified with phi29 to make DNA nanoballs (DNB) to encompass at least 300 copies of one molecule. The DNBs were loaded into the patterned nanoarray and paired end 100/150 bases reads were generated using combinatorial Probe-Anchor Synthesis (cPAS).

#### 2.4.2. Alignment and QC

Adapter removal and filtering of low-quality reads was done using SOAPnuke v2.1.2 (https://github.com/BGI-flexlab/SOAPnuke, accessed on 1 July 2020) using the following parameters: filter -n 0.001 -l 10 -A 0.25 -Q 2 -G 2 -f AAGTCGGAGGCCAAGCGGTCTTAGGAAGACAA -r AAGTCGGATCGTAGCCATGTCGTTCTGTGAGCCAAGGAGTTG. This was followed by the alignment of filtered reads to GRCh38 reference genome using the Burrows–Wheeler alignment MEM algorithm (bwa v0.7.15) [21]. The mapped SAM files were converted to BAM files using Samtools v1.5. Individual BAM files were merged using Picard Tools v2.8.1 MergeSamFiles (http://broadinstitute.github.io/picard/, accessed on 1 July 2020). Each BAM file was then processed using Picard Tools FixMateInformation and CleanSam commands. Next, duplicates were then marked and removed from individual BAM files using Picard Tools MarkDuplicates. Finally, to improve variant calling accuracy and recalibrate the base quality scores, the Genome Analysis Toolkit (GATK v4.0.3.0) [22] BaseRecalibrator and ApplyBQSR commands were applied. Coverage was calculate using Picard Tools CollectWgsMetrics. The recalibrated BAM files had a median coverage of 85–110× in tumour and 34–42× in normal samples (Appendix A, Appendix A).

#### 2.4.3. Variant Calling 

Three variant calling algorithms were used to call somatic variants: MuTect [23], Strelka2 [24] and MuSE [25].

#### 2.4.4. MuTect2 Pipeline

Variant calling was done using the steps outlined in GATK documentation for GATK v4.0.3 (https://gatk.broadinstitute.org/hc/en-us/articles/360035889791?id=11136; accessed on 1 July 2020). First, to infer cross-sample contamination, GetPileupSummaries and CalculateContamination steps were run for each tumour and matched with normal BAM files. Next, Mutect2 was run on all normal BAM files in tumour-only mode, and the normal calls were subsequently combined to generate a panel of normal (PoN) vcf files. For an increased coverage of common germline and artefactual variants, PoN contained three additional normal samples from lobular patients, which were not included in this study. Next, somatic variants were called in each tumour sample, using their matched normals, the panel of normals and a population germline variant resource from the Genome Aggregation Database (gnomAD) (http://gnomad.broadinstitute.org/, accessed on 1 July 2020) containing allele-specific frequencies. CollectSequencingArtifactMetrics was run to collect metrics on sequence context artefacts from each tumour sample, and its output was used to filter for the confident MuTect2 callset using FilterMutectCalls and FilterByOrientationBias, all using GATK v4.0.3. Variants that passed filters were selected using SelectVariants from GATK v4.1.8.1.

#### 2.4.5. Strelka2 Pipeline

Tumour and normal BAM files from previous steps were used to call snvs and indels, using Strelka2 v2.9.10 somatic workflow configuration. For best somatic indel performance, we ran Strelka2 in conjunction with Manta [26], using their default settings. After variant calling, vcftools v0.1.16 [27] was used to extract and keep only variants with the PASS filter.

#### 2.4.6. MuSE Pipeline

Somatic point mutations were called from our tumour and normal BAM files with MuSE v1.0rc default parameters for whole genome sequencing, dbSNP v146 vcf file and indexed reference genome GRCh38 as input. Only variants with PASS and Tiers1-4 based calls were kept for our final analysis.

#### 2.4.7. Annotations of Variants

All variants were annotated using annovar [28] (version 2020-06-08). In order to annotate Strelka2 vcf files, an in-house bespoke script was used to incorporate genotype fields into the vcf files to make them annovar compatible. The variants’ overlapping repeat regions were annotated by RepeatMasker track, downloaded from UCSC Table browser [29].

#### 2.4.8. Selecting High Confidence Calls

In order to select high confidence SNVs, only variants called by at least 2 out of 3 callers were retained. Indels shared between Mutect2 and Strelka2 calls were defined as high confidence indels.

#### 2.4.9. Post Hoc Variant Filtering

Consensus somatic SNVs and indels were subject to the following filters:(1)Remove multiallelic sites;(2)Exclude common variants with prevalence >1% in gnomAD database;(3)Remove variants with tumour allele frequency <0.01;(4)Filter for variants with depth <10 in both tumour and matched normal sample;(5)Keep variants with reads supporting alternate allele in tumour sample ≥ 5 and reads supporting the alternative allele in matched normal sample ≤ 2;(6)Remove synonymous variants.

#### 2.4.10. Identification of Recurrent Mutations

Using our consensus SNVs and Strelka2 indels (not limiting to high confidence indels), we generated a table of mutation counts per gene excluding variants in non-genic regions (intergenic, upstream and downstream). Gene annotations are based on annovar RefGene database definitions from the National Centre of Biotechnology Information (NCBI) [30]. For each gene, we estimated the proportion of samples carrying variants in ILC and IDC samples, and statistically quantified the difference in proportions between the two groups, using the proportion test. Only genes with an absolute difference ≥ 3 between each group were considered.

#### 2.4.11. Tumour Subclonal Deconvolution

Using our consensus filtered variants, the tumour heterogeneity analysis was conducted with MOBSTER R package v1.0.0 [31] in R statistical environment v4.0.3. The analysis was conducted using default fast run parameters (auto_setup = “FAST”). A single VAF column was generated from consensus calls, using the VAF (variant allele frequency) available from Mutect2; otherwise, Strelka2’s VAF was used.

#### 2.4.12. Copy Number Analysis

ascatNgs [32] was used to identify CNAs, using tumour and matched normal BAM files. Additionally, purity and ploidy were estimated using ascatNgs and sequenza (v3.0.0). Where ascatNgs could not estimate the correct aberrant cell fraction and ploidy, sequenza (v3.0.0) [33] estimates were used. Then, bedtools intersect [34] was used to annotate copy number segments with gene names, and a bespoke R script was used to pre-process ascatNgs output and aggregate CNA segments across genes. Where genes mapped to multiple CNA segments, the CNA with the largest overlap was kept. Effective copy number states were estimated by adjusting for tumour ploidy (Appendix A).

#### 2.4.13. Single-Base Substitutions (SBS) Signature Analysis

Inference of trinucleotide single-base substitutions signatures was performed, using the R (v4.1.0) package MutationalPatterns (v3.2.0). Representative single-base substitution signature profiles were downloaded from the COSMIC database: http://cancer.sanger.ac.uk/cancergenome/assets/signatures_probabilities.txt (accessed on 22 October 2021).

### 2.5. DNA Methylation Pre-Processing and Analysis

Methylation profiling was performed using Illumina Infinium MethylationEPIC BeadChip 850K arrays, using 1 μg of input tumour DNA. Samples were processed using the Infinium assay methylation protocol guide. Methylation data analysis was performed in R statistical environment v.3.6.0 using the cross-package Bioconductor workflow described by Maksimovic et al. [35]. R packages minfi (v1.32.0) was used for data pre-processing and quality control. The filtering process included the removal of probes containing detection *p*-values  > 0.05 in one or more samples, probes on the Y chromosome, probes carrying common SNPs at CpG sites and cross-reactive probes from Pidsley et al. [36]. The final data set contained 792,584 probes available for analysis (Appendix A).

Probes overlapping with the promoter of selected genes from the β-catenin pathway, *CDH3* and *CTNND1* were chosen. To identify probes exhibiting differential methylation between the two components of a given sample, probes were further reduced to those where the methylation status (methylated: beta value > 0.3, unmethylated: beta value ≤ 0.3) was opposite in the two components and the beta values had an absolute difference (delta) ≥ 0.1.

### 2.6. Data Availability

Whole genome sequencing data are publicly available through the EBI ENA repository under the accession number: PRJEB48274. SNVs (Data S1), indels (Data S2) and methylation data are publicly available at DOI: 10.5281/zenodo.5776310.

## 3. Results

### 3.1. Assessment of the E-Cadherin Pathway in E-Cadherin Negative and Aberrant/Positive Components of E-Cadherin Heterogeneous Breast Cancers

We identified 18 pathologically confirmed fresh frozen invasive lobular breast carcinomas of classical histology with heterogeneous E-cadherin expression (EcadhetILC), of which 11 were amenable to microdissection to separate out E-cadherin negative (Neg) and aberrant/positive components (Abr), where aberrant E-cadherin protein expression was defined as weak or cytoplasmic staining [9]. For nine cases, dissection of the separate components yielded sufficient DNA for genomic profiling (whole genome sequencing and methylation profiling) and passed quality control measures (see methods). All nine cases were ER positive and HER2 negative. Of these, eight cases showed aberrant E-cadherin protein expression in the Abr component; one tumour (ML10) harboured E-cadherin positive ductal carcinoma in situ (DCIS); and one tumour (ML2) had positive E-cadherin protein expression (Figure 1, Appendix A).

Evidence suggests that some mixed IC-NST and ILC breast cancers harbour differential E-cadherin (*CDH1*) mutations between the IC-NST and ILC components [9]. To ascertain whether this was also the case in EcadhetILC, we analysed the repertoire of *CDH1* mutations between the distinct components of EcadhetILC. Of the nine cases subjected to whole genome DNA sequencing (Appendix A, Appendix A), 5/9 (56%) harboured clonal loss of function mutations in *CDH1* (stopgain, frameshift deletion and frameshift insertion) (Appendix A). These were present in both dissected components (E-cadherin negative and E-cadherin aberrant) at equal frequencies of the EcadhetILC tumours, suggesting that mutation of *CDH1* is an early event in these tumours. The five *CDH1* mutant (*CDH1^MT^*) tumours also showed concomitant LOH of *CDH1* and copy number loss of the remaining allele (Appendix A, Appendix A). Of the four *CDH1* wild-type (*CDH1^WT^*) tumours, two (ML6 and ML12) showed differential copy number loss of the *CDH1* locus at 16q22.1 with the Neg component showing a copy number loss of *CDH1* and the Abr component retaining both alleles (Appendix A, Appendix A). Previous studies have reported sporadic cases of multiple cancer types with high DNA methylation levels at the *CDH1* promoter, suggesting epigenetic silencing as an alternative mechanism for the downregulation of *CDH1* [7,8]. To assess this in the different components of EcadhetILC tumours, we interrogated the methylation profiles detected from the methylation array profiling (Appendix A). In agreement with the recent TCGA study, we did not identify promoter methylation of *CDH1* in any of the tumours.

We next sought to address the heterogeneity in β-catenin and p120-catenin in both components of EcadhetILC (Appendix A, Appendix A). Consistent with the E-cadherin IHC results, the Neg components of EcadhetILC’s showed negative β-catenin stain and cytoplasmic p120-catenin [11,12,13]. The Abr components of the EcadhetILC’s 7/9 tumours showed negative β-catenin expression, with ML13 showing incomplete membranous stain and ML10 (DCIS showing positive protein expression). In concert with this, positive membranous p120-catenin was detected in ML10, whereas in all other tumours, the Abr component showed weak positive membranous and weak cytoplasmic expression, indicative of cytoplasmic relocation in line with a ILC phenotype.

We next assessed promoter methylation of β-catenin and p120-catenin and additional WNT pathway genes. No promoter methylation was observed in β-catenin (Appendix A). We did, however, identify promoter methylation of *CTNNA1* (α- catenin) in ML1_Neg (*CDH1*^MT^); *CTNNA2* (catenin alpha-2) and *APC* promoter methylation in ML10_Abr (*CDH1^WT^*); and placental (P)-cadherin (*CDH3*) promoter methylation of the *CDH1*^MT^ tumour ML8_Neg (Appendix A). Additionally, other members of the WNT signalling pathway also showed promoter methylation in ML10_Abr (*CDH1^WT^*) suggestive of aberrant WNT signalling in the DCIS lesion of this particular tumour.

We next sought to assess if somatic mutations in the promoter regions of *CDH1,* and α-, β- and γ-catenin could explain the heterogeneity of E-cadherin in these tumours lacking *CDH1* mutations. In both components of ML3, one SNV (C > T, VAF:_ML3_Neg = 0.391, ML3_Abr = 0.275) in *CTNNA1* was detected 3822 bp upstream of the transcription start site (TSS) within the first intron. Whilst the function of this non-coding mutation upstream of the TSS region of *CTNNA1* is unknown, this may highlight an additional inactivation mechanism of *CDH1* mediated through *CTNNA1* in the EcadhetILC tumours [37]. Of note, no copy number loss of *CTNNA1* was observed in any tumour.

Taken together, these results suggest that genomic alterations of *CDH1* may explain the differential pattern of E-cadherin protein expression of a subset of EcadhetILC tumours; however, in the majority of cases, the differences in E-cadherin expression could be due to additional mechanisms.

### 3.2. E-Cadherin Negative and Aberrant Components of EcadhetILC Are Clonally Related

In mixed IC-NST and ILCs, some tumours show divergence of the morphological components early during tumour evolution, where lobular morphology can derive from a ductal ancestor (where both DCIS and LCIS are present), or later during tumour progression (where only DCIS is detectable) [9]. However, the evolution in the context of EcadhetILC tumours is unknown. To assess this, we analysed the whole genome sequence data of the Neg and Abr components of the nine cases. We initially sought to determine if there was a difference in the overall mutational pattern of Neg and Abr components of the nine EcadhetILC. In general, we detected a moderate to high number of single nucleotide variants (SNVs, median overlap coefficient = 0.61) and insertions and deletions (indels, median overlap coefficient = 0.49) between both components in each tumour (Figure 2A,B, Data S1–2). ML10 and ML12 demonstrated poor overlap in the SNVs and indels of their respective components, highlighting that divergence in these two tumours occurred early on in tumour development. Next, we examined the relative fraction of six base substitutions considering the pyrimidines context (SNVs: C > A, C > G, C > T, T > A, T > C and T > G). The fractions were similar between both components of each tumour (Figure 2C), with C > T base substitutions being the most prevalent in line with observations in breast cancer in general [38].

### 3.3. E-Cadherin Negative and Aberrant/Positive Components of E-Cadherin Heterogeneous Breast Cancers Show Similar Driver Alterations

To determine if the repertoire of mutations would be distinct between the E-cadherin-negative and E-cadherin-positive/aberrant components of EcadhetILCs, and to identify potential driver mutations restricted to either one of the components, we next analysed the somatic copy number and mutational repertoire of both components of the nine cases that was subjected to whole genome sequencing. Focussing on known breast cancer driver genes [6], we identified clonal mutations in *PIK3CA* and *MAP2K4* in ML10_Abr, and a clonal mutation in *FRMPD2* in ML1_Abr (Figure 2D). Of note, we did not observe any differences in the clonal structure of the two components of these tumours, except for ML8_Neg, which showed the presence of two subclones (Appendix A). In particular, a subclonal mutation in *PIK3CA* (VAF = 0.102) and *CTCF* (VAF = 0.143) was identified in ML8_Neg (Figure 2E). Additionally, no recurrent alterations were identified that were commonly exclusive to either negative or aberrant component (with an observed prevalence in at least 3 samples), suggesting that copy number alterations occur early on in the development of EcadhetILCs, in agreement with other published studies in breast cancer [6,38,39].

To assess genome-wide mutational patterns, we summarised the whole genome mutational profiles to the 12 single-base substitution (SBS) signatures of breast cancer [38,39]. Using the six base substitutions in the pyrimidines context along with the flanking 5′ and 3′ bases, 96 classes of SBS were established and compared between the E-cadherin negative and aberrant components. Consistent with previous breast cancer genome profiling studies [38,39], SBS signatures 1 and 5 were prevalent across the majority of the samples regardless of the underlying component (median cosine similarity: SBS signature 1 = 0.75, SBS signature 5 = 0.78, Figure 3A). While SBS signature 1 is driven by C > T mutations in the N[C]G trinucleotide context and SBS signature 5 is driven by the presence of both C > T and T > C mutations, both are associated with age. Both components of ML8, ML3 and ML13 as well as sample ML1_Neg also exhibited high similarity with SBS signature 2, which is enriched with C > T mutations in the T[C]N context and may result from the activation of AID/APOBEC cytidine deaminases. SBS signature 2 along with SBS signature 13 are known to be associated with hypermutated breast cancers [40], which is consistent with our data showing higher mutation rates in both components of ML8, ML3 and sample ML1_Neg (Figure 2A). Interestingly, SBS signatures 1 and 2 were strikingly different between the E-cadherin negative and aberrant components of ML1 (Figure 3A). ML1_Neg demonstrated a strong presence of SBS signature 2 (cosine similarity = 0.97) while SBS signature 1 was enriched (cosine similarity = 0.8) in the ML1_Abr sample (Figure 3B), suggesting potentially distinct clonal progression in the two components.

## 4. Discussion

In this study, we provide the first and largest genome wide characterisation of ILCs harbouring heterogeneous E-cadherin expression. Overall, we find that EcadhetILC tumours harbour similar genomic alterations regardless of the E-cadherin protein expression and display genomic alterations found to be enriched in both ER+/Luminal-A IC-NST and ILC tumours. We additionally identify a proportion of tumours with driver gene mutations differential between two components, suggesting earlier divergence in these tumours. Overall, albeit lacking power, our data suggest that EcadhetILC belongs to the spectrum of lobular breast cancers.

Of the nine tumours studied here, four of these harboured clonal mutations in *CDH1* in both the Neg and Ab components. This contrasts with a recent small study assessing the mutational repertoire of mixed IC-NST/ILC’s (*n* = 4) that suggested *CDH1* mutations in MDL cancers are associated with clonal divergence [9]. However, the difference in E-cadherin protein expression could be explained in some tumours, due to a differential copy number loss of *CDH1* in cases that were *CDH1*^WT^. In line with Ciriello et al., we did not observe DNA methylation at the *CDH1* promoter in any of the nine tumours analysed [6]. Our results agree with a recent study that highlighted that ILCs can progress via a ‘ductal’ pathway of tumour progression [9]. Amongst the tumour where we analysed DCIS with strong E-cadherin protein expression, no *CDH1* mutations or promoter methylation were observed, suggesting that divergence from ductal morphology in the *in-situ* disease to a lobular morphology is due to additional mechanisms. Interestingly, in ML10 *CDH1^WT^* where the DCIS was profiled, promoter methylation of WNT pathway genes were detected in the Abr component, including *APC* and *CTNNA2* and may reflect the earlier divergence and underlying drivers of abrogated cell–cell adhesion in this tumour. Whilst promoter methylation of *APC* was previously reported in ILCs, there is no known direct link between promoter methylation and E-cadherin protein expression [8]. Of note, we identified somatic mutations in the promoter region and promoter methylation of *CTNNA1* (α- catenin) in two tumours. Somatic *CTNNA1* mutations were previously linked with MDL tumours and were shown to lead to atypical localisation of E-cadherin, lobular morphology and increased invasiveness of cells [37]. However, the mutations identified here were in the promoter region, and as such, the significance of these alterations remains to be elucidated.

Consistent with previous studies [38,39], all tumours, except ML1_Neg, showed typical breast cancer mutational profiles with enrichment of SBS signatures 1 and 5, confirming that the mutational repertoire of EcadhetILC tumours is similar to that of breast cancer regardless of IC-NST or the lobular morphology. Interestingly, ML1_Neg along with ML3, ML8 and ML13 showed enrichment of SBS signatures 2 and 13 which captures the activation of AID/APOBEC cytidine deaminases. This association is known to co-occur with a higher mutation rate [40], a phenomenon reproducible in ML1_Neg, ML3 and ML8. These tumours were also *CDH1* mutant, suggesting that the AID/APOBEC mutator phenotype is more likely to occur in tumours harbouring *CDH1* mutations and may be a precursor to the acquisition of a *CDH1* mutation.

Recent analysis from TCGA categorised tumours of mixed ductal and lobular histology into ILC-like, mainly harbouring *CDH1* mutations and alterations more commonly seen in pure ILCs (such as *FOXA1* and *TBX3* mutations) and IDC-like (lacking *CDH1* mutations and enrichment of mutations more commonly seen in pure IDC) [6]. In our series, we observed that a proportion of *CDH1*^MT^ tumours also harboured clonal mutations known to be enriched in pure IDCs (*MAP3K1*, *GATA3*) alongside ILC-enriched mutations (*TBX3*) present in both the Neg and Ab components, suggesting that these tumours inherently harbour genomic features that are found to be enriched in both pure IDC and ILC tumours. Although these tumours show differential E-cadherin expression, they show a typical lobular histological growth pattern, indicating that EcadhetILCs may not be classified as IC-NST- or ILC-like, but are more likely a hybrid where divergence has occurred at different stages of evolution in each individual tumour. Additionally, we included two samples with DCIS in our analysis, and although the E-cadherin-negative ILC component showed patterns of genomic alterations consistent with an earlier divergence from a common ancestor, they both lacked a *CDH1* mutation, indicating that they may harbour other mechanisms of E-cadherin inactivation.

Our study has a number of limitations, mainly, the small number of cases that were included in the study and also that only a proportion were amenable to dissection and thus analysed; however, it is the largest and most comprehensive sequencing study of mixed ILCs to date. A number of the EcadhetILCs that we identified from the tissue bank harboured complete intermixing of E-cadherin-negative and aberrant/positive cells and as such were excluded from this study. It would be of interest to use single-cell approaches to dissect the molecular differences between these two subpopulations and to ascertain whether divergence occurs later on in development. Although we identified differential promoter methylation in some of the tumours analysed between the two components, we were unable to look at the direct consequences of this, as we were unable to perform matched RNA profiling.

Taken together and akin to previous studies, our results show that the different phenotypic and morphological components of EcadhetILCs are clonally related and not derived from the collision of independent lesions [9]. Whilst the majority of the molecular alterations were found to be similar between the two components analysed, we identified mutations in driver genes, such as *PIK3CA*, *MAP2K4* and *FRMPD2,* that were distinct to one component. *PIK3CA* mutations are known to predict the clinical response to PI3K inhibitors [41,42], and, as such, the evaluation of phenotypically distinct components in these studies is important. Overall, our study contributes to the understanding of the molecular heterogeneity of these understudied tumours and highlights that EcadhetILCs should be considered as part of the spectrum of invasive lobular breast cancers.

## Figures and Tables

**Figure 1 cancers-14-00295-f001:**
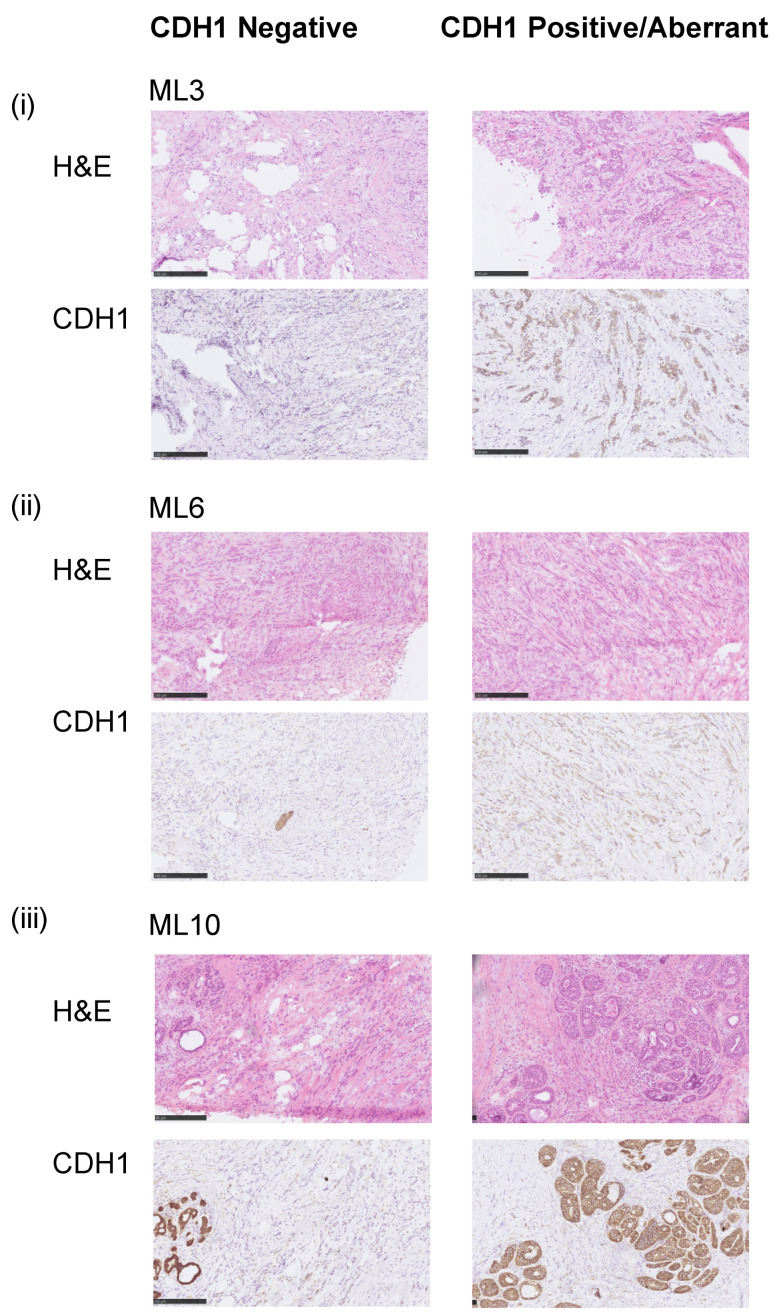
Representative samples of E-cadherin staining. Micrographs of representative hematoxylin and eosin (H&E) stained sections and E-cadherin immunohistochemistry (IHC) of three selected E-cadherin heterogeneous breast cancers included in this study (scale bar IHC, 200 μm). (**i**) ML3 *CDH1^MT^*, (**ii**) ML6 *CDH1^WT^* and (**iii**) ML10 *CDH1^WT^.* Note in ML10, the aberrant component is DCIS.

**Figure 2 cancers-14-00295-f002:**
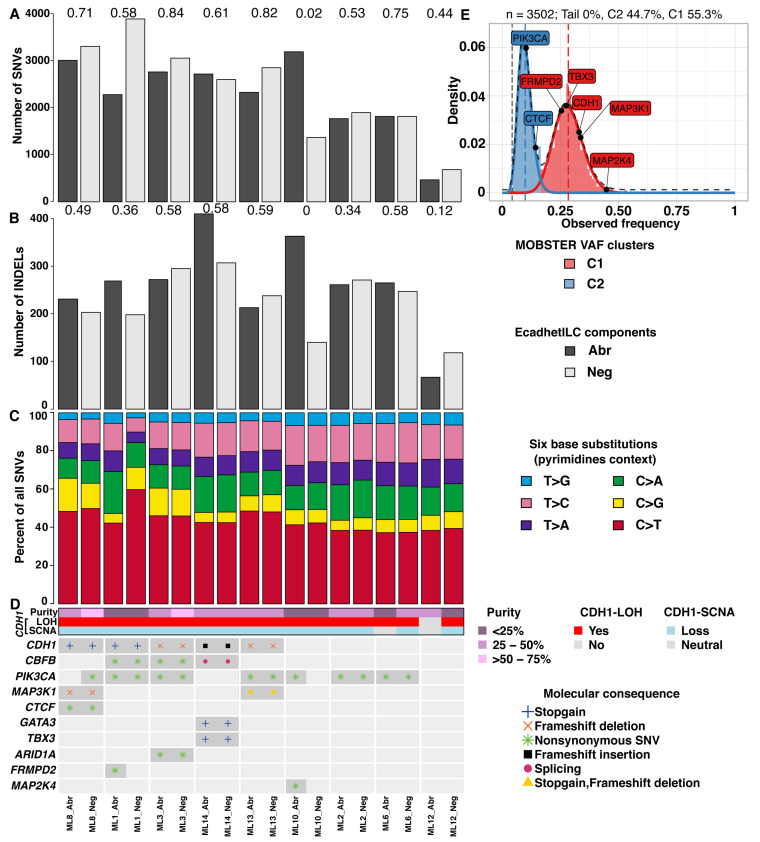
Summary of mutational analysis. (**A**) Bar plot of number of SNVs for each component of patient samples. (**B**) Bar plot of indels for each component of patient samples. (**C**) Proportion of SNVs split into six classes in pyrimidines context. (**D**) Heatmap of recurrently altered breast cancer genes [6] with at least one mutation found in our study. Numbers on bar plots show overlap coefficient of SNVs (**A**) and indels (**B**) between paired patient samples. Note that ML6_Abr has a ploidy estimate of 4, so is copy neutral with LOH of the *CDH1* locus. (**E**) MOBSTER-inferred clonal clusters of ML8_Neg showing two distinct VAF-derived distributions.

**Figure 3 cancers-14-00295-f003:**
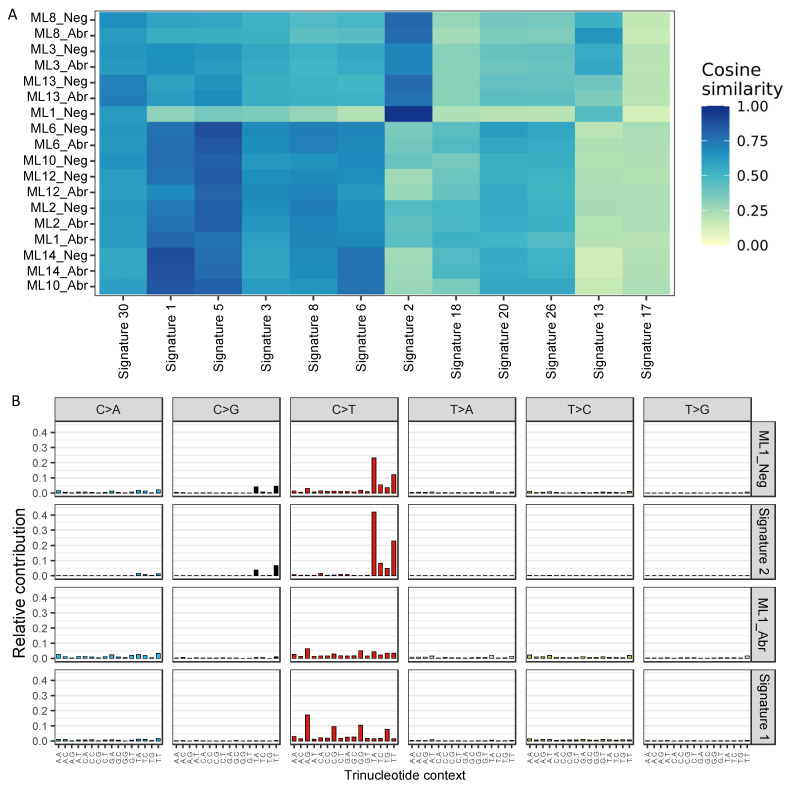
Mutational signatures (**A**) Heatmap of cosine similarities between mutational profiles and 12 breast cancer single-base substitution signatures [38]. Agglomerative hierarchical clustering was performed using complete linkage with Euclidean distance measure on both dimensions. (**B**) Single-base substitution signature profiles for ML1 components and the corresponding representative profiles from the COSMIC signature database.

## Data Availability

Raw targeted sequencing data have been deposited into the NCBI Sequence Read Archive under the accession PRJEB48274.

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
