# Peer review of "Assessment of the Molecular Heterogeneity of E-Cadherin Expression in Invasive Lobular Breast Cancer"

_cancers, 2022, doi:10.3390/cancers14020295_

Round 1

Reviewer 1 Report

Overall, this is a well-done study to dissect the genetic differences between E-cad neg and Ecadpos/aberrant regions of invasive lobular cancer. However, there are specific concerns that need to be addressed before publication.

  1. Ab is the standard abbreviation for antibody and should not be used to abbreviate aberrant/positive.
  2. Figure 1 legend states "2-3 cases" - be specific, ER, PR, and WT are defined in the figure legend but not used in the figure. Please indicate what ML# means in the figure legend.
  3. Representative images of microdissected tissue (i.e. after microdissection) should be shown as well.
  4. All data, not just WGS, must be made available (i.e. including methylation profiling)- supplementary figure 5 is cited as this data but is not actually this data
  5. The staining data for beta-catenin and p120 catenin should be shown as supplementary data.
  6. MDL needs to be defined

Author Response

Overall, this is a well-done study to dissect the genetic differences between E-cad neg and Ecadpos/aberrant regions of invasive lobular cancer. However, there are specific concerns that need to be addressed before publication.

We thank the reviewers for the positive assessment of our manuscript.

  1. Ab is the standard abbreviation for antibody and should not be used to abbreviate aberrant/positive.

We have now amended this throughout the text and figures to “Abr“ to avoid confusion

2. Figure 1 legend states "2-3 cases" - be specific, ER, PR, and WT are defined in the figure legend but not used in the figure. Please indicate what ML# means in the figure legend.

This has now been amended.

“Figure 1. Micrographs of representative hematoxylin and eosin (H&E) stained sections and E-Cadherin immunohistochemistry (IHC) of three selected E-Cadherin heterogeneous breast cancers included in this study (scale bar IHC, 200 μm). ML3 CDH1MUT, ML6 CDH1WT and ML10 CDH1WT.”

3. Representative images of microdissected tissue (i.e. after microdissection) should be shown as well.

We thank the reviewer for highlighting this and sincerely apologise for inclusion of the incorrect methodology in the methods section. In actual fact, these cases have not been subjected to microdissection but macrodissection. Once the area of interest was chosen on the H&E and adjacent E-cadherin IHC slide, we directly recovered the tumour fragment which remained frozen in the resin used to freeze the tissue section. Only cases where distinctly separated regions were identified were included in the study.  We did not recover the sample by scraping a slide as for a paraffin block, so we do not have a slide image after extraction of the area of interest. We have now updated the methods to include this detail.

Macrodissection and DNA extraction

Three-μm-thick representative sections of the snap-frozen blocks of histologically confirmed classical ILC’s with heterogeneous E-Cadherin expression (EcadhetILC) were stained with E-Cadherin to provide a reference. Cases with clearly separated components were macrodissected from the frozen tissue block with a scapel to recover the tumour fragment and to ensure enrichment of the tumour cell content as previously described [17]. Matched germline DNA was extracted from peripheral blood as previously described [18] or macrodissected from adjacent normal breast tissue for each case. To avoid the possibility of morphologically appearing non-neoplastic cells harbouring somatic mutations, we prioritized the dissection of stromal cells and avoided normal breast ducts and lobules. DNA was extracted using the DNeasy Blood and Tissue Kit (Qiagen, Germany), according to the manufacturers' guidelines. DNA quantity and quality were analysed with the Agilent Bioanalyzer (Agilent,) and on 0.8% agarose gel with SYBR™ Safe DNA Gel Stain (Invitrogen).”

4. All data, not just WGS, must be made available (i.e. including methylation profiling)- supplementary figure 5 is cited as this data but is not actually this data

The methylation data alongside the Supplementary Data files associated with the study have now been provided under the zenodo accession number 10.5281/zenodo.5776310.

5. The staining data for beta-catenin and p120 catenin should be shown as supplementary data.

Representative images have now been included as Supplementary Figure 4

6. MDL needs to be defined

This has now been rectified:

Page 2: “In particular mixed IC-NST and ILC tumours (MDL’s)…”

Reviewer 2 Report

In this study, the authors investigated invasive lobular breast carcinomas (ILCs) that have heterogeneous expression of E-cadherin (CDH1). They microdissected the different components of the tumors and performed whole-genome sequencing to search for single-nucleotide changes, other mutations, or changes to DNA methylation patterns. The authors discovered that, although some tumor components expressed different cancer-driving mutations depending on their CDH1 expression status, most tumors in the analysis showed similar genomic changes in the different components of the tumor. In addition, no changes in DNA methylation of the CDH1 promoter region was observed in this study. Although loss of CDH1 expression is definitive in invasive lobular breast cancer, the authors suggest that ILCs with heterogeneous E-cadherin expression may still belong to this family. But the current study is a small one, with only 9 samples analyzed, so the authors suggest this classification only contingent on further studies.

The authors also discovered that single base substitution signatures were typical for breast cancer and that some tumors exhibited signatures consistent with AID/APOBEC cytidine deaminases, suggesting that mutations driven by this enzyme may be the cause of CDH1 mutations in these tumors.

The study combines the thoroughness of whole-genome sequencing with the precision of microdissection of heterogeneous regions of single tumors to search for cellular molecular differences in this unique tumor type. The authors reasonably conclude that ILCs expressing CDH1 in a heterogeneous pattern are not substantially different than other similar tumor types. The underlying driving mutations (primarily WNT signaling pathway analyzed here) are similar to other breast cancers, and in most tumors in this study, were similar between regions of the tumors expressing CDH1 or not. This contributes to the body of knowledge about how these driving mutations interact with E-cadherin mutations/expression patterns in lobular breast carcinoma.

I see little to critique in this paper, except for some very minor comments about a table and a figure below:

Supplementary Table 1: Columns B and C have the same heading but different results. This needs to be clarified.

Figure 2E is hard to read. Can it be enlarged or the resolution enhanced?

Author Response

I see little to critique in this paper, except for some very minor comments about a table and a figure below:

We thank the reviewer for the positive assessment of our study.

Supplementary Table 1: Columns B and C have the same heading but different results. This needs to be clarified.

This has now been rectified to distinguish between the negative and aberrant components

Figure 2E is hard to read. Can it be enlarged or the resolution enhanced?

We think this was due to single pdf submission and therefore resulting in poor resolution. We have now submitted high-resolution figures separately, as well as increased the font sizes in Figure 2E.

Reviewer 3 Report

The article is very interesting and very good structurate. I suggest few minor language and editing corrections.

Point 1. The abstract needs to be modified because its not easy to follow and avoided abbreviators.

Point 2. Specify in the introduction part of how E-cadherin is regulation in many cancers. what are the results of these investigations, how are they correlated with malignancy of different molecular subtypes?

Author Response

The article is very interesting and very good structurate. I suggest few minor language and editing corrections.

We thank the reviewer for the positive comments

Point 1. The abstract needs to be modified because its not easy to follow and avoided abbreviators.

We have now modified the abstract to avoid abbreviations where possible

Abstract: Mutations and loss of E-cadherin protein expression define the vast majority of invasive lobular carcinomas. In a subset of these cases heterogeneous expression of E-cadherin is observed either as wild-type (strong membranous) expression or aberrant expression (cytoplasmic expression). However, it is unclear as to whether the two components would be driven by distinct genetic or epigenetic alterations. Here we used whole genome DNA sequencing and methylation array profiling of two separately dissected components of nine invasive lobular carcinomas with heterogeneous E-cadherin expression. E-cadherin negative and aberrant/positive components of E-cadherin heterogeneous tumours showed a similar mutational, copy number and promoter methylation repertoire, suggesting they arise from a common ancestor, as opposed to the collision of two independent tumours. We found the majority of E-cadherin heterogeneous tumours harboured CDH1 mutations in both the E-cadherin negative and aberrant/positive components together with somatic mutations in additional driver genes known to be enriched in both pure invasive carcinomas of no special type and invasive lobular breast cancers, whereas these were less commonly observed in CDH1 wild-type tumours. CDH1 mutant tumours also exhibited a higher mutation burden as well as increased presence of APOBEC-dependent mutational signatures 2 and 13 compared to CDH1 wild-type tumours. Together, our results suggest regardless of E-cadherin protein expression, tumours showing heterogeneous expression of E-cadherin should be considered as part of the spectrum of invasive lobular breast cancers.”

Point 2. Specify in the introduction part of how E-cadherin is regulation in many cancers. what are the results of these investigations, how are they correlated with malignancy of different molecular subtypes?

We have now added a couple of sentences to highlight this in the introduction:

“At the molecular level the characteristic discohesive hallmark of ILC is due to dysregulation of cell-cell adhesion, primarily driven by lack of E-cadherin (CDH1) protein expression observed in the majority of cases and is a discriminating feature of ILC as strong membranous expression of E-cadherin is observed in the majority of IC-NST’s [2]. Thus, loss of E-cadherin in ILC is thought to contribute to its pathogenesis and different meta-static patterns, in particular the higher prevalence of metastasis to serosal surfaces such as the ovary and peritoneum [1-2].”

Round 2

Reviewer 1 Report

The authors have addressed all my comments